# Health-Related Quality of Life of Patients Treated with Biological Agents and New Small-Molecule Drugs for Moderate to Severe Crohn’s Disease: A Systematic Review

**DOI:** 10.3390/jcm11133743

**Published:** 2022-06-28

**Authors:** Hasan Aladraj, Mohamed Abdulla, Salman Yousuf Guraya, Shaista Salman Guraya

**Affiliations:** 1School of Medicine, Royal College of Surgeons Ireland-Bahrain, RCSI-Medical University of Bahrain (MUB), Adliya P.O. Box 15503, Bahrain; 19201165@rcsi.com; 2Clinical Sciences Department, College of Medicine, University of Sharjah, Sharjah 27272, United Arab Emirates; salmanguraya@gmail.com (S.Y.G.); ssalman@rcsi.com (S.S.G.)

**Keywords:** Crohn’s disease, biologics, small-molecule drugs, health-related quality of life (HRQoL)

## Abstract

Crohn’s disease (CD) leads to a poor health-related quality of life (HRQoL). This review aimed to investigate the effect of biological agents and small-molecule drugs in improving the HRQoL of patients with moderate to severe CD. We adopted a systematic protocol to search PubMed and Cochrane Central Register of Controlled Trials (CENTRAL), which was supplemented with manual searches. Eligible studies were RCTs that matched the research objective based on population, intervention, comparison and outcomes. Studies in paediatric populations, reviews and conference abstracts were excluded. Covidence was used for screening and data extraction. We assessed all research findings using RoB2 and reported them narratively. We included 16 multicentre, multinational RCTs in this review. Of the 15 studies that compared the effect of an intervention to a placebo, 9 were induction studies and 6 investigated maintenance therapy. Of these, 13 studies showed a significant (*p* < 0.05) improvement in the HRQoL of patients with CD. One non-inferiority study compared the intervention with another active drug and favoured the intervention. This systematic review reported a substantial improvement in the HRQoL of patients with CD using biological agents and small-molecule drugs. These pharmaceutical substances have the potential to improve the HRQoL of patients with CD. However, further large clinical trials with long-term follow-up are essential to validate these findings.

## 1. Introduction

Crohn’s disease (CD) is a debilitating inflammatory disease that affects any part of the gastrointestinal tract, resulting in intestinal and systemic manifestations. It is a chronic disease characterised by alternating periods of disease relapse and remission. The chronic nature, early age of onset and incapacitating intestinal and systemic manifestations account for major social and financial stressors. Some distressing factors in patients with CD include frequent hospital visits, long-term medications with their side effects, bowel stenosis, possible surgical interventions and the fear of developing cancer [1,2]. A major burden on healthcare systems is related to the management of the CD-specific chronic internal and perianal fistulas, which need special attention in highly specialised colorectal surgery centres. The most dreadful complication of CD remains colorectal cancer, with a reported incidence of 746,000 cases (10.0% of the total cancer burden in men) and 614,000 cases (9.2% of the cancer incidence in women) [3,4].

HRQoL is a multidimensional concept which pertains to vitality, social energy and physical wellbeing [5,6,7,8]. To determine the effect of disease activity on HRQoL, several disease-specific HRQoL questionnaires have been used, such as the McMaster inflammatory bowel disease questionnaire (IBDQ) [9], the short IBDQ [10], the rating form of inflammatory bowel disease patient concerns (RFIPC) [11] and the sickness impact profile (SIP) [12]. All such tools measure specific elements of the HRQoL of patients with a focus on the certain characteristics of vitality and mental and social wellbeing.

To improve HRQoL, traditionally, the contemporary management of CD was driven by a progressive, stepwise therapeutic intensification with a re-review of the clinical response according to symptoms. This approach did not improve the long-term clinical outcomes in patients with CD [13], leading to the introduction of a “treat to target” CD management strategy, which guides physicians in the regular assessment of disease activity using objective clinical and biological outcome measures and subsequent treatment modifications [14]. The STRIDE-II initiative, an Update on the Selecting Therapeutic Targets in Inflammatory Bowel Disease (STRIDE) Initiative, confirmed that the restoration of QoL is the most important long-term treatment target in CD, irrespective of other objective markers of inflammation [15].

In the last two decades, biological agents (generally large, complex molecules manufactured by biotechnology[16]) have emerged as novel therapeutic agents for CD. Since the approval of infliximab in 1999, five other agents have been approved. These drugs work by inhibiting TNF-alpha, integrin-alpha4 or IL23/12p40 [17,18]. Due to a rising number of non-responders to treatment and a deeper understanding of the pathophysiological mechanisms of CD, new drugs are being developed to target IL23p19 and the JAK/STAT mechanisms or to regulate gut leukocyte trafficking [18]. As a primary outcome measure of therapy and a key factor of consideration for decision-makers, HRQoL has become a frequently measured outcome in clinical trials.

In 2009, a systematic review reported that the then-approved biologics (infliximab, adalimumab, certolizumab and natalizumab) demonstrated clinical improvement in the HRQoL of patients with inflammatory bowel disease (IBD) [19]. Since then, despite a staggering upsurge in CD management strategies and the availability of novel biological agents, there has been a scarcity of literature that could validate their efficacy using the best clinical evidence. Therefore, this systematic review aimed to evaluate the outcomes of the currently approved and promising in-development biological agents and small-molecule agents in improving the HRQoL of patients with moderate to severe CD.

## 2. Methods

### 2.1. Objective

Our review targeted studies of patients with moderate to severe CD, measured using a Crohn’s Disease Activity Index (CDAI) score of 221 to 450 points or equivalent, being treated with biological agents and small-molecule agents such as TNF-alpha, integrin-alpha4 or IL23/12p40 inhibitors or those regulating the JAK/STAT mechanism or gut leukocyte trafficking. We included studies that compared interventions with placebos or any other drug. The co-primary outcomes of this review were the number of patients achieving clinically meaningful improvements in HRQoL using the inflammatory bowel disease questionnaire (IBDQ) or the SF-36 questionnaires and the mean change in IBDQ total score or the physical component summary (PCS) and mental component summary (MCS) of the SF-36. Only studies that reported the targeted outcomes were included.

### 2.2. The HRQoL Scales

The IBDQ is the most frequently used disease-specific HRQoL tool [20]. The IBDQ is a 32-item questionnaire with 4 domains: bowel symptoms, systemic symptoms, emotional functioning and social functioning. The IBDQ total score is the sum of responses to all the items, which use a 7-point Likert scale grading system with 1 reflecting a severe problem and 7, no problem at all. The total score ranges between 32 (very poor HRQoL) and 224 (perfect HRQoL) [19,21].

The SF-36 is a generic HRQOL tool mainly used in IBD clinical trials [20]. The SF-36 has two summary components, the PCS and the MCS, derived from scores in eight individual scales (physical functioning, role—physical, bodily pain, general health, vitality, social functioning, role—emotional and mental health). A scale of 0 to 100 is used to score eight scales, with better HRQoL indicated by a higher score [19].

### 2.3. Inclusion and Exclusion Criteria

All double- or triple-blinded randomised controlled trials (RCTs) published in English that met the objective of our review were included. We excluded studies regarding adolescents and children (under 18 years of age). Conference proceedings, systematic reviews and non-English studies were also excluded.

### 2.4. Search Strategy

On 25 January 2022, a literature search, designed in conjunction with a senior librarian, was carried out on the databases of PubMed and Cochrane Central Register of Controlled Trials (CENTRAL). No limits were placed on the time span. Our search did not include grey literature. The capture–recapture method was used to verify the completeness of the search strategy results [22]. Keywords of Crohn’s disease, HRQoL, IBDQ, SF-36, anti-TNF and infliximab were used. To narrow our results towards RCTs, we used a search strategy suggested by the Cochrane handbook that is highly sensitive for identifying results of RCTs [23]. A manual search of the reference lists and www.clinicaltrials.gov (29 January 2022) was also conducted independently. Details of the search strategy are shown in Appendix B.

### 2.5. Data Extraction

The screening of titles, abstracts and full-text articles was conducted by two independent reviewers (H.A. and M.A.) using the Covidence software, using the defined inclusion and exclusion criteria as benchmarks. Any discrepancies were discussed and resolved by the two reviewers. The same software was used for the extraction of data. A customised template containing fields such as general information (title, study ID and registration number), the characteristics of the included studies (aim, date conducted and funding) and the results was used.

### 2.6. Risk of Bias Assessment

To ascertain the risk of bias, Cochrane collaboration’s risk of bias tool 2 (RoB2, 22 August 2019 version) was used independently by two reviewers (H.A. and M.A.) [24]. Any discrepancies were discussed, and then, a third researcher was consulted to secure a consensus. RoB2 is an outcome-based tool examining five domains which may lead to bias (bias arising from the randomisation process, deviations from intended interventions, missing outcome data, measurement of the outcome and the selection of the reported result). Studies that were rated high in one domain or raised some concerns in multiple domains that substantially lowered the confidence in the results were rated high overall. The risk of bias in relevant outcomes was reported, and those studies with a high risk of bias were not excluded based on those results.

### 2.7. Strategy for Data Synthesis

The extracted data were categorised according to the interventions used. The results were reported narratively using descriptive statistics, with the addition of tables and graphs where appropriate. If a study showed a statistically significant improvement (*p* < 0.05) in at least one dose group at the end of the study period, the intervention was considered to be effective in improving HRQoL.

This review was reported according to The Preferred Reporting Items for Systematic reviews and Meta-Analyses (PRISMA) guidelines [25]. This review is registered with The International Prospective Register of Systematic Reviews (PROSPERO), an open-access online database of systematic review protocols, with the registration number CRD42022306394 [26].

## 3. Results

Our first search retrieved 306 and 303 records from PubMed and CENTRAL, respectively. A further 38 records were retrieved by hand searching www.clinicaltrails.gov and reference lists. After the removal of 95 duplicates, 552 title/abstracts were screened, which showed 433 irrelevant reports. Furthermore, 44 reports did not have retrievable full-text articles, 9 were ongoing studies and 44 were excluded for other reasons, as depicted in the flowchart. Finally, 22 reports of 16 studies were included (Figure 1). Four studies met the inclusion criteria. However, we excluded those studies as there was no data about HRQoL outcomes [27,28,29,30].

### 3.1. Characteristics of the Included Studies

Our systematic review identified 16 studies. Fifteen compared their investigated interventions to a placebo. Only SONIC compared its intervention (infliximab) to another active drug (azathioprine) [31]. All included studies were multinational, multicentre RCTs. The total number of participants in this review was 7463 and ranged between 108 and 1281. Three studies investigated infliximab [31,32,33], three studies investigated certolizumab pegol [34,35,36], three studies investigated ustekinumab [37], two studies investigated natalizumab [38,39] and one study each investigated adalimumab [40], filgotinib [41], upadacitinib [42], tofacitinib [43] and apilimod mesylate [44]. Nine were induction studies, and six were maintenance studies. SONIC was an induction study with a maintenance extension [31].

A clinically meaningful improvement (MCID) in HRQoL is defined as an increase of ≥16 points in the IBDQ total score and an increase of 3 to 5 points in the SF-36 PCS and MCS scores. [45] Based on these values, nine studies defined MCID in the IBDQ as ≥16 points. Three studies defined MCID in the SF-36 PCS and MCS as ≥5 points. Only the PRECiSE 2 trial defined MCID in the PCS and MCS as 4.1 and 3.9 points, respectively [35].

### 3.2. Risk of Bias

A total of 157 outcomes were assessed. Of these, 104 (66%), 1 (0.6%) and 52 (33%) were rated as having high risks, some concerns or low risks of bias overall, respectively. As many as 9.5% of outcomes [33,44] were rated has having some concerns in domain 1: randomisation process. Meanwhile, 22.3% of outcomes [38,39,43] were rated as having a high risk in domain 2: deviations from intended interventions. A total of 88 (56%) [32,33,34,35,37,39,40,44] outcomes were rated as having a high risk in missing outcome data (domain 3), while 14% [36,37] of outcomes were rated as having some concerns. In domain 5: selection of the reported results, 38 (24.2%) [32,33,34,40] and 41 outcomes (26.1%) [33,35,39] were rated as having a high risk and some concerns, respectively. All outcomes were rated as having low risks in domain 4 (Table 1 and Figure 2).

All studies used random allocation sequences except ACCENT I [33] and Sands et al. [44]. Their outcomes raised some concerns in domain 1, as the allocation sequence was concealed, and the baseline characteristics were consistent with randomisation. All studies used the intention to treat (ITT) or modified intention to treat (mITT) populations for analysis, except ENCORE [38], ENACT-2 [39] and Tofacitinib [43], in which there was no information available about the analysed population. Hence, their outcomes were rated as high risk in domain 2. The outcomes of PRECiSE 1 [34] and PRECiSE 2 [35] (more withdrawals in the placebo groups) and Targan et al. [32], ACCENT I [33], CHARM [40], IM UNITI [37], ENACT-2 [39] and Sands et al. [44] (lack of information about the number of or reason for withdrawals) were rated as high risk in domain 3. The major reason for withdrawal in Rutgeerts et al. [36], UNITI I and II [37] and Filgotinib [41] was a lack of efficacy. However, their outcomes showed some concerns, as the number of withdrawn participants was balanced between the study groups. In UNITI I and II, [37] unlike the IBDQ outcomes (which had low risk), PCS and MCS outcomes were rated as having some concerns due to missing outcome data. The protocols of PRECiSE- 2 [35] and ENACT-2 [39] (in which there were some concerns that results were selected), Targan et al. [32], CHARM [40], PRECISE 1 [34], Rutgeerts et al. [36] and Tofacitinib [43] (in which it was likely that results were selected; high risk of bias) were not found. The IBDQ results were likely selected (high risk) in ACCENT I [33], but selection was not suspected in PCS and MCS results (some concerns). The full details of the RoB2 assessment can be found in Appendix A.

### 3.3. Effect of Interventions on HRQoL

The effect of interventions on HRQoL is summarised in Table 2 for the SONIC [31] study and Table 3 for the placebo-controlled trials. The tables include the study ID/registration number, intervention, dosage, results and conclusion.

#### 3.3.1. Infliximab vs. Azathioprine

SONIC [31] compared infliximab with azathioprine. In the induction period, the difference in the mean change in the IBDQ total score in the infliximab group was significantly higher than the azathioprine group at weeks 2, 18 and 26 (*p* < 0.05), but not at weeks 6 and 10 (*p* = 0.10). In the maintenance phase, the difference was statistically significant at weeks 34 and 42 but not at week 50 (*p* = 0.001, *p* = 0.04 and *p* = 0.09, respectively).

#### 3.3.2. Infliximab vs. Placebo

Two studies, Targan et al. [32] and ACCENT I [33], compared infliximab with placebo.

Targan et al. [32] compared three groups using 5, 10 or 15 mg/kg infliximab induction with a placebo. Patients had a statistically higher mean IBDQ score in all infliximab groups at week 4 (*p* < 0.05, compared to placebo).

ACCENT 1 [33] examined the effect of two infliximab maintenance regimens, 5 mg/kg or 10 mg/kg infliximab, following a 5 mg/kg three-dose induction and compared them with a single dose of 5 mg/kg induction followed by a placebo. At week 10, the three-dose group had a higher mean IBDQ score compared to the single-dose induction group (*p* < 0.05). Higher IBDQ scores were maintained for both maintenance groups (5 mg/kg and 10 mg/kg infliximab) at week 30 (*p* < 0.05 and *p* < 0.01) and week 50 (*p* < 0.05 and *p* < 0.001), respectively, compared to the single-dose induction group. Up to week 14, all treatment groups had an increase exceeding the MCID. Following week 14, the infliximab maintenance groups maintained this increase, while it decreased to below 16 points in the induction-only group. The PCS scores were significantly greater (*p* < 0.05) for both maintenance groups at weeks 10, 30 and 52 compared to the single-dose induction group. The difference in MCS scores was only significant at week 54, comparing the 10mg/kg maintenance group with the single-dose group (*p* < 0.05).

#### 3.3.3. Adalimumab vs. Placebo

The CHARM trial compared adalimumab maintenance, 40 mg every other week or weekly, with adalimumab induction only (placebo maintenance) [40].

Following a significant increase of 44.3 points (*p* < 0.0001, week 4 vs. baseline) in the mean IBDQ in the open-label induction phase, IBDQ scores continued to increase in the adalimumab maintenance groups (approximately 5 points), while IBDQ scores deteriorated in the induction-only group. There were statistically significant differences in the mean IBDQ total scores at all visits after week 4 between adalimumab maintenance groups and the induction-only group (*p* < 0.001 for adalimumab every other week and *p* < 0.05 for adalimumab weekly). After a year of maintenance (at week 56), patients in the adalimumab group had an IBDQ score of 18 points higher than those in the placebo group, a difference that exceeded the MCID of 16 points.

The differences in PCS scores were statistically significant at all visits following week 4 in the adalimumab-every-other-week maintenance group compared to the induction-only group (*p* < 0.05), while differences in the MCS were only significant at week 56 (*p* < 0.05). In total, 77% of adalimumab-every-other-week patients achieved an MCID of ≥5 points in the PCS compared to 61% in the induction-only group (*p* < 0.01). In the MCS, improvement was achieved by 67% and 54% of adalimumab-every-other-week and placebo patients, respectively (*p* < 0.05). Differences in the mean PCS and MCS between the adalimumab-weekly group and the placebo group were not statistically significant.

#### 3.3.4. Certolizumab Pegol vs. Placebo

Three studies compared certolizumab pegol and a placebo. One study [36,46] had four arms comparing certolizumab (100 mg), certolizumab (200 mg) or certolizumab (400 mg) with a placebo. The PRECiSE 1 study had two groups comparing 400 mg of certolizumab with a placebo (administered at weeks 0, 2 and 4 and then every 4 weeks) [34]. In PRECiSE 2 [35,47], following an open-label induction of 400 mg of certolizumab at weeks 0, 2 and 4, patients received either maintenance certolizumab (400 mg) or a placebo.

Rutgeerts et al. and Schreiber et al. [36,46] reported statistically significant changes in the mean IBDQ at all reported timepoints for the 400 mg group compared to the placebo group, with the greatest change at week 10 (certolizumab pegol (400 mg): 32.2 points vs. 18.6 points for placebo; *p* ≤ 0.05). The 200 mg group had significant changes at weeks 2 and 4 compared to the placebo group (*p* ≤ 0.05), while changes in the 100 mg group were not statistically significant. Differences in the mean IBDQ between the certolizumab pegol and placebo arms were statistically significant at week 26 in both PRECiSE 1 and PRECiSE 2 (*p* = 0.03 and *p* < 0.001, respectively). PRECiSE 2 also reported significant differences in the IBDQ means at week 16 (*p* = 0.008). The percentages of patients achieving an MCID in the IBDQ at week 26 were significantly greater in the certolizumab groups in both PRECiSE 1 and 2 (*p* = 0.01 and *p* < 0.001, respectively) compared to the placebo groups.

Only PRECiSE 2 used the SF-36 tool for the estimation of HRQoL. Patients in the certolizumab group showed statistically significant (*p* < 0.05) differences at week 26 in the mean change and proportion achieving an MCID compared to the placebo group.

#### 3.3.5. Ustekinumab vs. Placebo

The UNITI trials compared ustekinumab and a placebo [37,48]. UNITI I and UNITI II induction studies compared a single intravenous infusion of 130 mg of ustekinumab or 6 mg/kg ustekinumab to a placebo. Patients had an inadequate response or intolerance to tumour necrosis factor (TNF) antagonists (UNITI I) or conventional therapy (UNITI II). Patients with a clinical response were re-randomised to maintenance therapy with subcutaneous ustekinumab (90 mg) every 12 weeks (q12w) or every 8 weeks (q8w) for 44 weeks and compared to the placebo in IM UNITI.

In both induction studies, the mean change and proportion of patients achieving an MCID in the IBDQ total score in both ustekinumab groups were statistically significant at week 8 compared to the placebo groups (*p* < 0.05). In the maintenance study at week 20, the mean decrease from the maintenance baseline was significantly less in the q12w group but not in the q8w group compared to the placebo group (*p* = 0.035 and *p* = 0.183, respectively). The mean decrease at week 44 was significantly less in both ustekinumab maintenance groups (*p* < 0.001 and *p* = 0.003, q12w and q8w compared to the placebo group, respectively). A significantly greater proportion of patients achieved MCIDs in the IBDQ in the ustekinumab (q8w) but not the ustekinumab (q12w) group (*p* = 0.014 and *p* = 0.140, respectively, compared to the placebo group).

In UNITI II, the mean change from baseline in the PCS and MCS scores was significant for both ustekinumab doses at week 8 compared to the placebo dose (*p* < 0.05). In UNITI I, the only significant change at week 8 in the mean score was in the MCS of the ustekinumab 6 mg/kg group compared to the placebo group (*p* = 0.006). The same pattern was seen in MCID proportions, significant (*p* < 0.05) in UNITI II for the MCS and PCS in both doses but only significant in the MCS for the 6 mg/kg group in UNITI I.

In the maintenance study at week 44, the mean decrease in the PCS and MCS from the maintenance baseline was significantly less in the ustekinumab (q8w) group compared to the placebo group (*p* < 0.01), while it was only significantly less in the MCS in the ustekinumab (q12w) group compared to the placebo group (*p* < 0.05). Changes in the means of MCS and PCS were not significant at week 20 for both groups. Both groups had significantly (*p* < 0.05) higher proportions of patients with MCID improvements at week 44 in the PCS and MCS, except for the PCS in the q12w group.

#### 3.3.6. Natalizumab vs. Placebo

Two studies compared natalizumab and a placebo. The ENCORE trial compared natalizumab as induction therapy to a placebo [38,49]. The ENACT-2 trial compared maintenance natalizumab with a placebo in patients who responded to natalizumab induction in ENACT-1 [39].

Induction treatment with natalizumab in the ENCORE trial showed a statistically significant (*p* < 0.001) increase in the mean IBDQ total score and the mean PCS score (compared to the placebo) but not in the MCS (*p* = 0.052).

Maintenance natalizumab in ENACT-2 maintained increases in the mean IBDQ total score, PCS and MCS scores achieved from the induction therapy in ENACT-1. The decrease from the change achieved in week 12 (randomisation of ENACT-2) was significantly less in the natalizumab group compared to the placebo group (*p* < 0.01) for all subsequent weeks in the IBDQ total score and PCS. MCS scores were not significant at weeks 24 and 36 but reached significance at weeks 48 and 60 (*p* < 0.01 and *p* < 0.001, respectively, compared to the placebo). The proportion of patients with MCIDs was significantly greater at weeks 36, 48 and 60 in the IBDQ and MCS and at all weeks in the PCS in the natalizumab group (*p* < 0.05, compared to the placebo).

#### 3.3.7. Filgotinib vs. Placebo

The FITZROY study compared oral filgotinib to a placebo [41]. There was a 16-point difference favouring the filgotinib group compared to the placebo in the mean change from baseline of the IBDQ total score. This difference was statistically significant (*p* = 0.0046) and clinically meaningful.

#### 3.3.8. Upadacitinib vs. Placebo

The CELEST study compared five doses of oral upadacitinib (3 mg, 6 mg, 12 mg or 24 mg twice daily or 24 mg once daily) with a placebo as induction therapy [42,50]. Changes in the mean IBDQ total score at weeks 8 and 16 were only statistically significant in the 6 mg and 24 mg twice-daily groups (*p* ≤ 0.05, compared to the placebo). A significantly greater proportion of patients achieved clinically meaningful improvement in the IBDQ in all upadacitinib groups at week 16 and only the 6 mg twice-daily group at week 8 (*p* ≤ 0.05, compared to placebo).

#### 3.3.9. Tofacitinib vs. Placebo

One study compared three doses of tofacitinib (5 mg, 10 mg or 15 mg twice daily) with a placebo [43]. The 15 mg arm was closed early after the enrolment of only 16 participants. Therefore, this arm was not included in the efficacy analysis. Statistical significance was not calculated for HRQoL outcomes, and thus we could not determine the implications of the clinical evidence.

#### 3.3.10. Apilimod Mesylate vs. Placebo

Sands et al. compared two doses of apilimod mesylate (50 mg and 100 mg daily) as induction therapy with a placebo [44]. No statistically significant differences were found between either of the apilimod groups and the placebo at both time points (*p* > 0.3).

## 4. Discussion

The overarching goal of treatment for moderate to severe CD using biological agents and small-molecule drugs is the achievement of clinical remission and the arrestment or stabilisation of chronic intestinal inflammation. Our systematic review reported evidence-based clinical data from 16 RCTs and endorsed a superior role of biological agents and small-molecule drugs in improving the HRQoL outcomes in patients with CD. Out of the 16 studies identified in this systematic review, 15 studies compared their investigated interventions to a placebo. Only SONIC compared its intervention (infliximab) to another active drug (azathioprine) and favoured the intervention in the induction phase [31]. Excluding the SONIC study (as it had a different comparator) and one study [43], which did not report the statistical significance. A total of 8/14 studies used the intervention as induction therapy. In contrast, the remaining six RCTs studied maintenance therapy. All studies reported a significant difference in the mean change in the total IBDQ score, favouring the intervention group by the end of the study in at least one dose group, except for Sands et al. [44], who did not report a statistically significant difference. Essentially, three out of eight induction studies and five out of six maintenance studies reported mean changes in the PCS and MCS. Of the induction studies, two out of three studies showed a significant difference in the mean change in the PCS and MCS favouring the intervention group, while all maintenance studies had a significantly greater change in the mean PCS and MCS in their intervention groups.

In our systematic review, two induction and four maintenance studies reported the proportion of patients achieving MCIDs in the IBDQ, and all reported significantly higher proportions in the intervention groups. The only induction studies that reported MCIDs in the PCS and MCS were UNITI I and UNITI II. UNITI I had a significantly higher proportion of patients with MCIDs in the MCS, while UNITI II had a higher proportion in both the MCS and PCS [37]. Furthermore, two out of six maintenance studies reported MCIDs in the PCS and MCS. Both studies had significant findings in favour of the intervention group.

In the systematic review conducted by Vogelaar et al., the researchers found that biologics (infliximab, adalimumab, certolizumab and natalizumab) improved HRQoL [19]. Comparably, the findings of our systematic review are in corroboration with Vogelaar et al. and showed significant improvements in HRQoL using other biologics (ustekinumab) and small-molecule drugs (upadacitinib and filgotinib). A Cochrane review of biologics in ulcerative colitis (another form of inflammatory bowel disease) found that infliximab and adalimumab significantly improved HRQoL [45]. This review argued that the studies on CD have also shown significant improvement in HRQoL. Another systematic review has reported that adalimumab improved fatigue, an aspect of HRQoL [51].

Our systematic review could not measure all of the relevant HRQoL outcomes, including the proportion of patients achieving MCIDs. Due to the missing data and inconsistency in the results from the analysed studies, appropriate statistical analyses could not be used. Some interventions showed inconsistencies in the improvement between physical and mental aspects of HRQoL. Lastly, most studies did not report both co-primary outcomes in both the IBDQ and SF-36 PCS and MCS summary scales. Despite these shortcomings, this systematic review diligently provided valuable data from RCTs which scientifically proves the efficacy of biological agents and small-molecule drugs in improving the HRQoL outcomes in patients with moderate to severe CD.

## 5. Limitations

There are some limitations to this review. The search strategy was conducted on two databases and only on English-language articles. Several studies that may have affected the results of this review were excluded because they did not report the targeted outcomes, or they were ongoing studies, including a trial for vedolizumab (a common biologic in current use). Effect measures were not calculated, nor were statistical analyses, including a meta-analysis conducted. Thus, the overall effect was not calculated. Owing to inconsistent clinical data from some of the selected studies, the possibility of unintentional research bias cannot be excluded. Nevertheless, during the systematic review process, for the accuracy and verification of results, the researchers arranged periodic meetings for mutual discussions, data cross-verifications and consensuses.

## 6. Conclusions

The cutting-edge advancements in drug research and biotechnology have introduced novel biologics and small-molecule drugs for the treatment of CD. Our systematic review demonstrated clear evidence of the efficacy of biological agents and small-molecule drugs in improving HRQoL outcomes in patients with moderate to severe CD. Due to the paucity of the comparative analysis of biologics and small-molecule drugs with other agents in the published literature, this study may potentially guide physicians in positioning and relocating drugs in management algorithms for patients with CD.

## Figures and Tables

**Figure 1 jcm-11-03743-f001:**
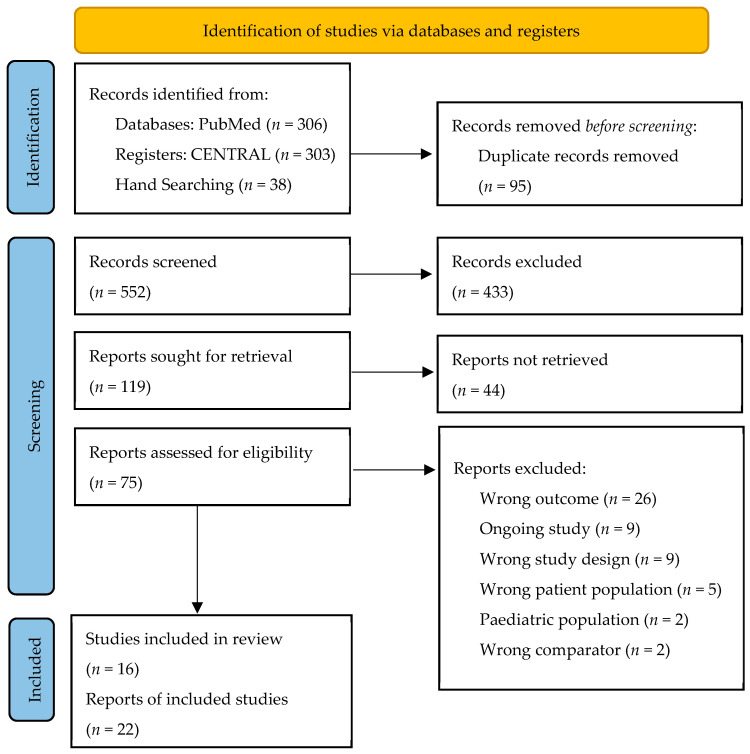
PRISMA flowchart for the stepwise screening and final selection of studies.

**Figure 2 jcm-11-03743-f002:**
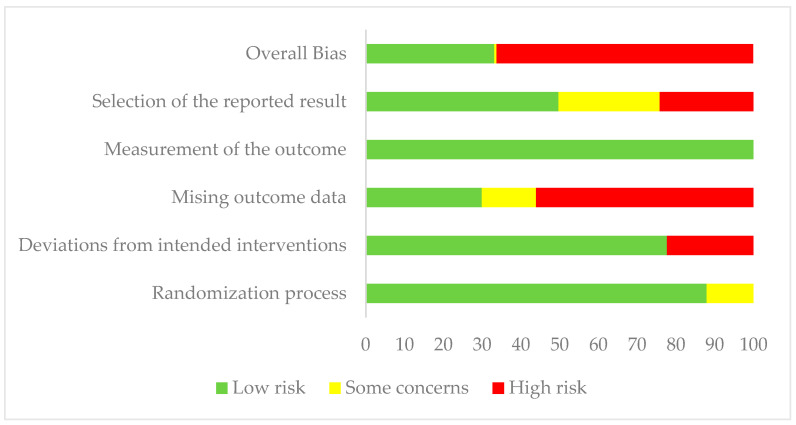
Risk of bias in the included study outcomes.

**Table 1 jcm-11-03743-t001:** The estimated risk of bias in the studies recruited for this systematic review (*n* = 16).

Study ID	Experimental	No. of Participants	GroupFavoured	Outcome	Weight	Risk of Bias
D1	D2	D3	D4	D5	O
SONIC [31]	Infliximab	508	Intervention	All HRQoL outcomes	8						
Targan et al. [32]	CA2	108	Intervention	All HRQoL outcomes	3						
ACCENT I [33]	Infliximab	573	Intervention	All IBDQ outcomes	5	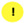					
All PCS and MCS outcomes	10	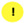				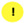	
CHARM [40]	Adalimumab	499	Intervention	All HRQoL outcomes	15						
PRECiSE 1 [34]	Certolizumab pegol	662	Intervention	All HRQoL outcomes	2						
PRECiSE 2 [35]	Certolizumab pegol	428	Intervention	All HRQoL outcomes	7					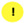	
Rutgeerts et al. [36]	Certolizumab pegol	292	Intervention	All HRQoL outcomes	5			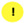			
UNITI I and II [37]	Ustekinumab	742 and 628	Intervention	All IBDQ outcomes	8						
All PCS and MCS outcomes	16			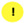			
IM UNITI [37]	Ustekinumab	1281	Intervention	All HRQoL outcomes	18						
ENCORE [38]	Natalizumab	509	Intervention	All HRQoL outcomes	3						
ENACT 2 [39]	Natalizumab	339	Intervention	All HRQoL outcomes	24					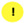	
FITZROY [41]	Filgotinib	174	Intervention	All HRQoL outcomes	1			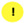			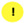
CELEST [42]	Upadacitinib	220	Intervention	All HRQoL outcomes	20						
Tofacitinib [43]	Tofacitinib	280	Not reported	All HRQoL outcomes	8						
Sands et al. [44]	Apilimod mesylate	220	Not significant	All HRQoL outcomes	4	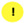					

D1: randomisation process, D2: deviations from the intended interventions, D3: missing outcome data, D4: measurement of the outcome, D5: selection of the reported result. 

 low risk, 
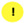
 some concerns, 

 high risk.

**Table 2 jcm-11-03743-t002:** The summary of findings for the SONIC [31] study. Results of the SONIC trials. All values are in means (SD).

Study Id and Registration Number	Intervention and Comparator	Dosage and Frequency	Results for	Intervention	Azathioprine	*p* Value	Conclusion
SONIC [31] NCT00094458	Infliximab (IV) and azathioprine (oral)	Infliximab 5mg/kg given at week 2,4,6 and then every 8 weeks OR azathioprine 2.5 mg/kg	**Mean IBDQ at baseline**	126.7 (30.3)	128 (29)	-	Non-inferiority trial favoured infliximab over azathioprine in improving HRQoL.
			Change at week 2	27.2 (26.1)	20.1 (24.3)	0.007	
			Change at week 6	34.8 (31.8)	28.3 (31.3)	0.10	
			Change at week 10	37.8 (35.6)	31.0 (31.7)	0.10	
			Change at week 18	39.9 (34.2)	30.3 (33.9)	0.01	
			Change at week 26	39.9 (36.6)	31.4 (35.4)	0.05	
			Change at week 32 *	55.8 (33.6)	39.1 (32.9)	0.001	
			Change at week 42 *	51.4 (32.8)	40.3 (32.1)	0.04	
			Change at week 50 *	51.6 (32.9)	43.0 (33.4)	0.09	

* SONIC extension for maintenance therapy.

**Table 3 jcm-11-03743-t003:** Summary of findings of the placebo-controlled studies (*n* = 15).

Study Id and Registration Number	Intervention	Dosage and Frequency	Results for	Intervention	Placebo	*p* Value	Conclusion
Targan et al. [32] NCT00269854	Infliximab (IV)	5 mg/kg single dose	**Mean IBDQ at baseline**	122 (29)	128 (29)	-	Infliximab significantly improved IBDQ in the short term.
			Mean at week 4	168 (36)	133 (28)	<0.001	
		10 mg/kg single dose	**Mean IBDQ at baseline**	116 (23)	128 (29)	-	
			Mean at week 4	146 (41)	133 (28)	0.02	
		15 mg/kg single dose	**Mean IBDQ at baseline**	118 (28)	128 (29)	-	
			Mean at week 4	149 (35)	133 (28)	0.03	
ACCENT I [33] NCT00207662	Infliximab (IV)	5 mg/kg every 8 weeks	**Mean IBDQ at baseline**	170 (26)	170 (29)	-	Both doses of infliximab maintenance maintained a significant increase in mean IBDQ and PCS at all time points. Mean MCS difference was not significant except at week 50 in the 10 mg/kg dose.
			Change at week 10	37.8	28.9	<0.05	
			Change at week 30	27.1	14.0	<0.05	
			Change at week 50	22.1	8.9	<0.05	
			**Mean PCS at baseline**	NR	NR	-	
			Change at week 10	8.6	4.9	<0.001	
			Change at week 30	7.3	3.1	<0.01	
			Change at week 50	6.1	2.5	<0.05	
			**Mean MCS at baseline**	NR	NR	-	
			Change at week 10	6.5	3.8	≥0.05	
			Change at week 30	4.6	2.9	≥0.05	
			Change at week 50	5.1	2.0	≥0.05	
		10 mg/kg every 8 weeks	**Mean IBDQ at baseline**	168 (31)	170 (29)	-	
			Change at week 30	31.7	14	<0.01	
			Change at week 50	30.2	8.9	<0.001	
			**Mean PCS at baseline**	NR	NR	-	
			Change at week 30	7.3	3.1	<0.01	
			Change at week 50	7.2	2.5	<0.01	
			**Mean MCS at baseline**	NR	NR	-	
			Change at week 30	4.9	2.9	≥0.05	
			Change at week 50	5.8	2.0	<0.05	
CHARM [40] NCT00077779	Adalimumab (SC)	40 mg every other week	**Mean IBDQ at baseline**	NR	NR	-	Both doses of adalimumab maintained a significant increase in mean IBDQ at week 56. Adalimumab every other week maintained a significant increase in mean PCS. Increase in mean MCS reached significance at week 56 in the every other week group. Adalimumab weekly showed no significance in SF-36 outcomes compared to placebo.
			Mean at week 56	176	NR	<0.001	
			**Mean PCS at baseline**	37.1 (7.9)	36.8 (8.0)	-	
			Mean at week 4 (OL)	44.5 (7.8)	44.3 (8.9)	-	
			Mean at week 12	46.9 (8.6)	44.5 (9.0)	<0.01	
			Mean at week 26	47.4 (9.2)	44.7 (8.6)	<0.01	
			Mean at week 56	47.5 (8.5)	45.3 (8.6)	<0.01	
			**Mean MCS at baseline**	38.2 (11.0)	38.6 (10.9)	-	
			Mean at week 4 (OL)	46.2 (10.4)	47.4 (10.4)	-	
			Mean at week 12	48.4 (10.7)	46.2 (11.0)	NS	
			Mean at week 26	48.2(10.6)	45.8 (11.4)	NS	
			Mean at week 56	48.7 (10.5)	45.9 (11.2)	<0.05	
			**PCS MCID **** at week 56	77	61	<0.01	
			**MSC MCID **** at week 56	67	54	<0.05	
		40 mg weekly	**Mean IBDQ at baseline**	NR	NR	-	
			Mean at week 56	171	NR	<0.05	
			**Mean PCS at baseline**	36.9 (9.6)	36.8 (8.0)	-	
			Mean at week 4 (OL)	43.7 (8.4)	44.3 (8.9)	-	
			Mean at week 12	46.0 (8.6)	44.5 (9.0)	NS	
			Mean at week 26	46.1 (8.7)	44.7 (8.6)	NS	
			Mean at week 56	47.1 (9.4)	45.3 (8.6)	NS	
			**Mean MCS at baseline**	36.3 (10.5)	38.6 (10.9)	-	
			Mean at week 4 (OL)	45.7 (9.3)	47.4 (10.4)	-	
			Mean at week 12	46.1 (11.9)	46.2 (11.0)	NS	
			Mean at week 26	46.1 (11.8)	45.8 (11.4)	NS	
			Mean at week 56	46.5 (12.4)	45.9 (11.2)	NS	
PRECiSE 1 [34] NCT00152490	Certolizumab pegol (SC)	400 mg every 4 weeks	**Mean IBDQ at baseline**	NR	NR	-	Certolizumab maintained significant increases in mean IBDQ, PCS and MCS after 26 weeks.
			Change at week 26	26.4 (35.1)	20.5 (33.1)	0.03	
			**IBDQ MCID *** at week 26	42.0	33.0	0.01	
PRECiSE 2 [35] NCT00152425	Certolizumab pegol (SC)	400 mg every 4 weeks	**Mean IBDQ at baseline**	NR	NR	-	
			Mean at week 16	170.0	162	0.008	
			Mean at week 26	175.7 (29.94)	167.9 (23.19)	<0.001	
			**Mean PCS at baseline**	NR	NR	-	
			Mean at week 26	48.1 (8.17)	46.4 (7.69)	0.014	
			**Mean MCS at baseline**	NR	NR	-	
			Mean at week 26	46.9 (11.53)	45.2 (11.83)	0.001	
			**IBDQ MCID *** at week 26	60.6	42.9	<0.001	
			**PCS MCID _a_** at week 26	51.2	33.8	<0.001	
			**MCS MCID _b_** at week 26	44.2	32.4	0.016	
Rutgeerts et al. [36]	Certolizumab pegol (SC)	100 mg every 4 weeks	**Mean IBDQ at baseline**	132.2 (30.60)	122.9 (26.60)	-	Induction with certolizumab (400 mg) significantly improved IBDQ.
			Change at week 2	16.6	10.6	NS	
		200 mg every 4 weeks	**Mean IBDQ at baseline**	122.9 (27.07)	122.9(26.60)	-	
			Change at week 2	21.8	10.6	<0.05	
		400 mg every 4 weeks	**Mean IBDQ at baseline**	126.5(25.20)	122.9(26.60)	-	
			Change at week 2	22.8	10.6	<0.05	
			Change at week 10	32.2	18.6	<0.05	
			Mean at week 12	156.4 (37.36)	140.5 (35.88)	<0.05	
			**IBDQ MCID *** at week 2	52.8	NR	NR	
			**IBDQ MCID *** at week 12	66.7	NR	NR	
UNITI I [37] NCT01369329	Ustekinumab (IV)	130 mg single infusion	**Mean IBDQ at baseline**	119.5 (29.47)	120.0 (29.27)	-	Both doses of induction ustekinumab significantly increased mean IBDQ in patients who previously failed treatment with TNF-alpha inhibitors. In this population, ustekinumab showed no significance in improving PCS. Only 6 mg/kg ustekinumab significantly improved MCS.
			Change at week 8	18.1 (28.02)	11.9 (26.51)	<0.05	
			**Mean PCS at baseline**	37.8 (7.12)	37.8 (7.12)	-	
			Change at week 8	3.2 (6.43)	2.6 (6.50)	NS	
			**Mean MCS at baseline**	37.3 (9.98)	37.8 (10.64)	-	
			Change at week 8	3.3 (9.41)	2.2 (8.47)	NS	
			**IBDQ MCID *** at week 8	46.9	36.5	0.019	
			**PCS MCID **** at week 8	33.3	30	NS	
			**MCS MCID **** at week 8	36.4	30	NS	
		6 mg/kg single infusion	**Mean IBDQ at baseline**	118.2 (26.64)	120.0 (29.27)	-	
			Change at week 8	22.1(28.59)	11.9 (26.51)	<0.001	
			**Mean PCS at baseline**	37.2 (7.09)	37.8 (7.12)	-	
			Change at week 8	3.6 (6.75)	2.6 (6.50)	NS	
			**Mean MCS at baseline**	36.4 (9.89)	37.8 (10.64)	-	
			Change at week 8	4.9(9.28)	2.2 (8.47)	0.006	
			**IBDQ MCID *** at week 8	54.8	36.5	<0.001	
			**PCS MCID **** at week 8	34.9	30	NS	
			**MCS MCID **** at week 8	42.4	30	0.007	
UNITI II [37] NCT01369342	Ustekinumab (IV)	130 mg single infusion	**Mean IBDQ at baseline**	118.2 (30.99)	122.7 (31.32)	-	Both doses of induction ustekinumab significantly increased mean IBDQ, PCS and MCS in patients who previously failed conventional treatment. Ustekinumab groups had a significantly higher proportion of patients achieving MCID.
			Change at week 8	29.1 (33.82)	29.1 (33.82)	<0.001	
			**Mean PCS at baseline**	38.9 (7.62)	39.7 (7.19)	-	
			Change at week 8	5.1 (7.24)	2.6 (5.88)	<0.010	
			**Mean MCS at baseline**	37.2 (10.81)	37.1 (10.75)	-	
			Change at week 8	5.9 (10.55)	3.3 (9.47)	<0.010	
			**IBDQ MCID *** at week 8	58.7	41.1	<0.001	
			**PCS MCID **** at week 8	44	31.2	0.009	
			**MCS MCID **** at week 8	49.2	38.6	0.036	
		6 mg/kg single infusion	**Mean IBDQ at baseline**	122.8 (31.62)	122.7 (31.32)	-	
			Change at week 8	35.3 (36.05)	14.7 (26.96)	<0.001	
			**Mean PCS at baseline**	38.9 (7.05)	39.7 (7.19)	-	
			Change at week 8	6.0 (7.70)	2.6 (5.88)	<0.001	
			**Mean MCS at baseline**	37.9 (11.15)	37.1 (10.75)	-	
			Change at week 8	6.8 (11.34)	3.3 (9.47)	<0.001	
			**IBDQ MCID *** at week 8	68.1	41.1	<0.001	
			**PCS MCID **** at week 8	49.2	31.2	<0.001	
			**MCS MCID **** at week 8	51.3	38.6	0.014	
IM UNITI [37] NCT01369355	Ustekinumab (SC)	90 mg every 12 weeks (q12w)	**Mean IBDQ at baseline**	165.8 (32.82)	163.6 (31.76)	-	By the end of the study, ustekinumab (q8w) maintained significant improvement across all outcomes. The q12w group maintained improvement in mean IBDQ and MCS.
			Change at week 20	−6.3 (37.04)	−12.8 (34.05)	0.035	
			Change at week 44	−8.9 (43.08)	−21.5 (39.26)	<0.001	
			**Mean PCS at baseline**	47.1 (8.10)	46.3 (8.21)	-	
			Change at week 20	−1.7 (7.18)	−1.7 (7.67)	NS	
			Change at week 44	−2.3 (9.31)	−3.6 (9.33)	NS	
			**Mean MCS at baseline**	46.4 (10.66)	45.7 (10.89)	-	
			Change at week 20	−1.3 (11.53)	−2.7 (10.78)	NS	
			Change at week 44	−1.9 (12.68)	−4.4 (11.06)	<0.050	
			**IBDQ MCID *** at week 44	61.3	50.4	NS	
			**PCS MCID **** at week 44	41.7	34.7	NS	
			**MCS MCID **** at week 44	46.7	28.9	0.005	
		90 mg every 8 weeks (q8w)	**Mean IBDQ at baseline**	170.5 (29.33	163.6 (31.76)	-	
			Change at week 20	−8.9 (31.46)	−12.8 (34.05)	NS	
			Change at week 44	−9.9 (34.83)	−21.5 (39.26)	<0.010	
			**Mean PCS at baseline**	47.4 (7.52)	46.3 (8.21)	-	
			Change at week 20	−0.6 (6.37)	−1.7 (7.67)	NS	
			Change at week 44	−0.9 (7.14)	−3.6 (9.33)	<0.010	
			**Mean MCS at baseline**	47.3 (9.91)	45.7 (10.89)	-	
			Change at week 20	−1.7 (9.01)	−2.7 (10.78)	NS	
			Change at week 44	−1.7 (9.76)	−4.4 (11.06)	<0.010	
			**IBDQ MCID *** at week 44	67.9	50.4	0.014	
			**PCS MCID **** at week 44	52.1	34.7	0.008	
			**MCS MCID **** at week 44	47.9	28.9	0.003	
ENCORE [38] NCT00078611	Natalizumab (IV)	300 mg every 4 weeks	**Mean IBDQ at baseline**	123.6 (31.06)	122.5 (28.44)	-	Natalizumab induction significantly increased mean IBDQ and PCS but not MCS.
			Change at week 12	26.7 (32.34)	15.2 (28.92)	<0.001	
			**Mean PCS at baseline**	34.7 (8.7)	34.6 (8.4)	-	
			Change at week 12	5.8 (8.2)	2.7 (6.7)	<0.001	
			**Mean MCS at baseline**	37.9 (10.8)	4.9 (10.5)	-	
			Change at week 12	4.9 (10.5)	2.6 (9.6)	0.052	
ENACT-2 [39] NCT00032786	Natalizumab (IV)	300 mg every 4 weeks	**Mean IBDQ at ENACT 1 baseline**	125 (30)	121 (30)	-	By week 60, the natalizumab group had a higher mean and proportion of patients achieving MCID in all three outcomes. Increase in mean IBDQ and PCS was significant at all timepoints, while mean MCS reached significance at week 48.
			Change at the end of ENACT-1	58.6 (30.0)	NG	-	
			Change at week 24	51.6 (31.4)	43.8 (35.0)	<0.01	
			Change at week 36	53.4 (32.9)	39.4 (38.9)	<0.001	
			Change at week 48	52.3 (32.9)	36.5 (39.6)	<0.001	
			Change at week 60	53.9 (33.6)	35.5 (40.3)	<0.001	
			**MCID *** at the end of ENACT-1	93.2	NG	-	
			**IBDQ MCID *** at week 24	85.8	77.9	NS	
			**IBDQ MCID *** at week 36	87.5	70.5	<0.01	
			**IBDQ MCID *** at week 48	86.7	65.1	<0.001	
			**IBDQ MCID *** at week 60	86.7	65.1	<0.001	
			**Mean PCS at ENACT 1 baseline**	33 (8)	34 (8)	-	
			Change at the end of ENACT-1	12.5(8.5)	NG	-	
			Change at week 24	12.5 (8.5)	8.8 (8.9)	<0.01	
			Change at week 36	13.4 (9.4)	7.6 (9.6)	<0.001	
			Change at week 48	12.9 (9.4)	6.9 (9.2)	<0.001	
			Change at week 60	12.6 (9.4)	6.8 (9.5)	<0.001	
			**PCS MCID **** at the end of ENACT-1	78.5	NG	-	
			**PCS MCID **** at week 24	79.1	63.0	<0.05	
			**PCS MCID **** at week 36	82.4	54.8	<0.001	
			**PCS MCID **** at week 48	77.9	53.4	<0.001	
			**PCS MCID **** at week 60	75.7	53.4	<0.001	
			**Mean MCS at ENACT 1 baseline**	39 (11)	37 (11)	-	
			Change at the end of ENACT-1	10.5 (10.5)	NG	-	
			Change at week 24	8.6 (10.5)	8.0 (11.0)	NS	
			Change at week 36	8.3 (11.4)	7.2 (11.0)	NS	
			Change at week 48	8.9 (10.2)	6.3 (12.1)	<0.01	
			Change at week 60	9.7 (10.5)	6.8 (12.4)	<0.001	
			**MCS MCID **** at the end of ENACT-1	68.7	NG	-	
			**MCS MCID **** at week 24	59.7	62.6	NS	
			**MCS MCID **** at week 36	61.0	53.0	<0.05	
			**MCS MCID **** at week 48	61.8	53.4	<0.001	
			**MCS MCID **** at week 60	64.7	52.6	<0.001	
FITZROY [41] NCT02048618	Filgotinib (oral)	200 mg once daily	**Mean IBDQ at baseline**	123.0 (2.8)	120.8 (3.6)	-	Filgotinib significantly improved mean IBDQ.
			Change at week 10	33.8 (3.0)	17.6 (5.1)	0.0046	
CELEST [42] NCT02365649	Upadacitinib (oral)	3 mg twice daily (BID)	**IBDQ mean at baseline**	115.2 (27.5)	118.0 (28.5)	-	Upadacitinib (6 mg and 24 mg BID) significantly increased mean IBDQ. By the end of the study, all doses had a higher proportion of patients achieving MCID in IBDQ compared to placebo.
			Change at week 8	19	17	NS	
			Change at week 16	21	13	NS	
			**IBDQ MCID *** at week 8	41	38	NS	
			**IBDQ MCID *** at week 16	46	24	≤0.05	
		6 mg twice daily (BID)	**IBDQ mean at baseline**	113.7 (25.9)	118.0 (28.5)	-	
			Change at week 8	35	17	≤0.05	
			Change at week 16	39	13	≤0.01	
			**IBDQ MCID *** at week 8	62	38	≤0.05	
			**IBDQ MCID *** at week 16	57	24	≤0.01	
		12 mg twice daily (BID)	**IBDQ mean at baseline**	115.2 (36.1)	118.0 (28.5)	-	
			Change at week 8	25	17	NS	
			Change at week 16	27	13	≤0.1	
			**IBDQ MCID *** at week 8	58	38	≤0.1	
			**IBDQ MCID *** at week 16	50	24	≤0.05	
		24 mg twice daily (BID)	**IBDQ mean at baseline**	113.8 (36.0)	118.0 (28.5)	-	
			Change at week 8	40	17	≤0.01	
			Change at week 16	41	13	≤0.01	
			**IBDQ MCID *** at week 8	53	38	NS	
			**IBDQ MCID *** at week 16	56	24	≤0.01	
		24 mg once daily (QID)	**IBDQ mean at baseline**	120.7 (36.3)	118.0 (28.5)	-	
			Change at week 8	23	17	NS	
			Change at week 16	22	13	NS	
			**IBDQ MCID *** at week 8	57	38	≤0.1	
			**IBDQ MCID *** at week 16	49	24	≤0.05	
Tofacitinib [43] NCT01393626	Tofacitinib (oral)	5 mg twice daily	**Mean IBDQ at baseline**	117.89 (27.98)	118.50 (28.48)	-	P-values were not reported; we cannot determine the significance of the results.
			Change at week 8	41.20 (3.90)	26.58 (3.76)	NR	
			**Mean PCS at baseline**	38.49 (6.78)	37.12 (7.66)	-	
			Change at week 8	8.07 (0.956)	3.72 (0.927)	NR	
			**Mean MCS at baseline**	34.85 (11.68)	36.50 (12.26)	-	
			Change at week 8	7.88 (1.178)	6.47 (1.143)	NR	
			**IBDQ MCID *** at week 8	75	61.4	NR	
		10 mg twice daily	**Mean IBDQ at baseline**	113.67 (28.45)	118.50(28.48)	-	
			Change at week 8	40.05 (3.90)	26.58 (3.76)	NR	
			**Mean PCS at baseline**	35.28 (8.49)	37.12 (7.66)	-	
			Change at week 8	7.28 (0.967)	3.72 (0.927)	NR	
			**Mean MCS at baseline**	35.84 (10.68)	36.50 (12.26)	-	
			Change at week 8	7.13 (1.180)	6.47 (1.143)	NR	
			**IBDQ MCID *** at week 8	76.5	61.4	NR	
Sands et al. [44] NCT00138840	Apilimod mesylate (oral)	50 mg daily	**Mean IBDQ at baseline**	NR	NR	-	There were no significant differences between the intervention and placebo groups.
			Change at day 29	17.0	18.7	0.76	
			Change at day 43	17.7	23.2	0.33	
		100 mg daily	**Mean IBDQ at baseline**	NR	NR	-	
			Change at day 29	17.0	18.7	0.73	
			Change at day 43	19.5	23.2	0.48	

NR: not reported, NG: the drug was not administered, IV: intravenous, SC: subcutaneous, IBDQ: inflammatory bowel disease questionnaire, PCS: physical component summary, MCS: mental component summary. A clinically meaning full improvement (MCID) was defined as an increase of ≥ 16 points in the IBDQ score and an increase of 3–5 points in SF-36 PCS and MCS. All mean change values are in mean (SD). All MCID values are percentages (%). MCID *: proportion of patients achieving an improvement of ≥ 16 in the IBDQ score. MCID **: proportion of patients achieving an improvement of ^3^5 points in the SF-36 PCS or MCS score. MCID _a_: proportion of patients achieving an improvement of ≥ 4.1 points in the SF-36 PCS score. MCID _b_: proportion of patients achieving an improvement of ≥3.9 points in the SF-36 MCS score.

## Data Availability

The data presented in this study are available in this article.

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
