# Peer review of "Health-Related Quality of Life of Patients Treated with Biological Agents and New Small-Molecule Drugs for Moderate to Severe Crohn’s Disease: A Systematic Review"

_jcm, 2022, doi:10.3390/jcm11133743_

Round 1

Reviewer 1 Report

This study reports the improvement in health-related quality of life in patients with Crohn's disease using biologics/small molecules. The authors did an extensive review; however, from the reader's viewpoint, some data should be mentioned. Please find my comments below. 

1. There is no report about vedolizumab, one of the common biologic agents used currently. Please note the information in the discussion. 

2. It would be interesting to do a network meta-analysis. Paschos et al did a systematic review and network meta-analysis in ulcerative colitis (Aliment Pharmacol Ther 2018, DOI: 10.1111/apt.15005).  

3. This study includes only randomized controlled trials without any observational studies possibly because the inclusion criteria are restricted to the studies reporting complete outcomes. However, the real-world data is also interesting.  

4. The presentation of the results in table form is a bit difficult to understand. Please try to present in graphs. 

Author Response

Thank you for the review, please find out our responses to the comments below.

This study reports the improvement in health-related quality of life in patients with Crohn's disease using biologics/small molecules. The authors did an extensive review; however, from the reader's viewpoint, some data should be mentioned. Please find my comments below. 

  1. There is no report about vedolizumab, one of the common biologic agents used currently. Please note the information in the discussion. 

Noted, line 584. It was also referenced with 3 other reports which met the inclusion criteria but did not report HRQoL outcomes, line 243.

  1. It would be interesting to do a network meta-analysis. Paschos et al did a systematic review and network meta-analysis in ulcerative colitis (Aliment Pharmacol Ther 2018,DOI: 10.1111/apt.15005).  

Thanks for the comment. The referred systematic review aimed to “compare the impact of interventions for moderate-to-severe UC on health-related quality of life (HRQL)”. The study has concluded that “induction treatment with infliximab, adalimumab, golimumab, vedolizumab or tofacitinib improves quality of life compared to placebo”. In sharp contrast, our systematic review aimed to evaluate the outcomes of the currently approved and promising in-development biological agents and small molecule agents in improving HRQoL in patients with moderate to severe CD. The overarching purpose was to evaluate the efficacy of the drugs and not to compare drugs. Therefore, the studies under consideration are different and hence the study design and outcomes.

  1. This study includes only randomized controlled trials without any observational studies possibly because the inclusion criteria are restricted to the studies reporting complete outcomes. However, the real-world data is also interesting.  

The randomised controlled trails (RCTs) are the most reliable evidence to report the efficacy of interventions as the rigorous process of these trails guarantee a very low research bias and minimize the risk of confounding factors that would potentially influence the results. The element of randomization provides a rigorous tool to examine cause-effect relationships between an intervention and outcome. Wessely S. Randomised controlled trials: The gold standard. Evidence in the psychological therapies: A critical guide for practitioners. 2001:46-59.We used RCTs in our systematic review as they provide best clinical evidence in a specific subject and the inference drawn from the findings of RCTs are the most reliable and accurate body of scientific knowledge.

  1. The presentation of the results in table form is a bit difficult to understand. Please try to present in graphs. 

Thank you for the comment. We have added a description of the table content for more clarity, line 318.

Reviewer 2 Report

This is an interesting and relevant narrative synthesis on quality-of-life improvement after biological agents and small molecules treatment for patients with moderate to severe Crohn´s disease.

Introduction

Please, include fistulas and stenosis as important factors that affect quality of life of patients with Crohn´s disease.

Please, update information about STRIDE consensus to STRIDE II. It is important to mention that this consensus have already made recommendations about quality of life as a new treatment goal.

Turner D, Ricciuto A, Lewis A, D'Amico F, Dhaliwal J, Griffiths AM, Bettenworth D, Sandborn WJ, Sands BE, Reinisch W, Schölmerich J, Bemelman W, Danese S, Mary JY, Rubin D, Colombel JF, Peyrin-Biroulet L, Dotan I, Abreu MT, Dignass A; International Organization for the Study of IBD. STRIDE-II: An Update on the Selecting Therapeutic Targets in Inflammatory Bowel Disease (STRIDE) Initiative of the International Organization for the Study of IBD (IOIBD): Determining Therapeutic Goals for Treat-to-Target strategies in IBD. Gastroenterology. 2021 Apr;160(5):1570-1583.

Methods

Please, correct CDAI 220 to 450 > change 221 to 450

Results

Did you find any study with vedolizumab?

Is it possible to calculate effect sizes excluding studies with incomplete data? It would be interesting to compare the impact of each drug and to rank them in a network meta-analysis. If this is not possible, please, change title to narrative synthesis, instead of systematic review.

Serious shortcomings in reviews that use “narrative synthesis” have been identified, including unclear links between the included data, the synthesis, and the conclusions. Please, use Synthesis Without Meta-analysis (SWiM) guideline in addition to PRISMA to present your data if a meta-analysis is not possible.

Line 207 > All studies used random allocation sequences expect ACCENT I… Except instead of expect?

Line 210 >  intension to treat… intention to treat.

Author Response

Thank you for the review, please find out our responses to the comments below.

This is an interesting and relevant narrative synthesis on quality-of-life improvement after biological agents and small molecules treatment for patients with moderate to severe Crohn´s disease. 

 Introduction

Please, include fistulas and stenosis as important factors that affect quality of life of patients with Crohn´s disease. 

Thank you. We have added more details about Crohn’s disease specific stenosis and fistulas.

Please, update information about STRIDE consensus to STRIDE II. It is important to mention that this consensus have already made recommendations about quality of life as a new treatment goal.

Turner D, Ricciuto A, Lewis A, D'Amico F, Dhaliwal J, Griffiths AM, Bettenworth D, Sandborn WJ, Sands BE, Reinisch W, Schölmerich J, Bemelman W, Danese S, Mary JY, Rubin D, Colombel JF, Peyrin-Biroulet L, Dotan I, Abreu MT, Dignass A; International Organization for the Study of IBD. STRIDE-II: An Update on the Selecting Therapeutic Targets in Inflammatory Bowel Disease (STRIDE) Initiative of the International Organization for the Study of IBD (IOIBD): Determining Therapeutic Goals for Treat-to-Target strategies in IBD. Gastroenterology. 2021 Apr;160(5):1570-1583.

 Thank you. We have updated the information, line 78.

Methods

Please, correct CDAI 220 to 450 > change 221 to 450

The CDAI range has been changed to 221 to 450.

Results

Did you find any study with vedolizumab?

Yes, we did find a relevant study, but it did not report HRQoL data.

This information has been added to the discussion, line 584. It was also referenced with 3 other reports which met the inclusion criteria but did not report HRQoL outcomes, line 243.

Is it possible to calculate effect sizes excluding studies with incomplete data? It would be interesting to compare the impact of each drug and to rank them in a network meta-analysis. If this is not possible, please, change title to narrative synthesis, instead of systematic review.

Serious shortcomings in reviews that use “narrative synthesis” have been identified, including unclear links between the included data, the synthesis, and the conclusions. Please, use Synthesis Without Meta-analysis (SWiM) guideline in addition to PRISMA to present your data if a meta-analysis is not possible.

Thanks for the comment that has been taken from the following link in its entirety.

https://www.bmj.com/content/368/bmj.l6890. The reference is the publication Research Methods & Reporting Synthesis without meta-analysis (SWiM) in systematic reviews: reporting guideline.

However, the observation is not relevant to our systematic review. Table 3 illustrates a detailed summary of the key findings of our systematic review including study ID and registration, intervention (drug), dosage and frequency, placebo, p values and main conclusion of each study. The data provides strong evidence about the impact of each intervention as determined by the p values.  Thus, the conclusions are not based on narrative synthesis but on concrete statistical results. Therefore, the PRISMA guidelines are the right fit for our systematic review.

Line 207 > All studies used random allocation sequences expect ACCENT I… Except instead of expect?

Thank you, we have corrected in the text.

Line 210 > intension to treat… intention to treat.

Thank you, we have corrected in the text.

Reviewer 3 Report

In this article "Health related quality of life of patients treated with biological agents and new small-molecule drugs for moderate to severe  Crohn's disease: a systematic review" the authors looked at quality of life in biologics. Unfortunately this is not a new topic. extensive grammar editing is required for the paper. 

1. the introduction needs to be shortened. there is too much information that may be redundant and as a result may sway the reader away from the main goal at hand.

2. the results section should be made more succinct .

3. Given that the data from this paper is qualitative data and not quantitative, there is for sure methodical errors that play a role and affect the results. Were you able to account for the error given that some data may be missing from the studies you extracted

Author Response

Thank you for the review, please find out our responses to the comments below.

In this article "Health related quality of life of patients treated with biological agents and new small-molecule drugs for moderate to severe  Crohn's disease: a systematic review" the authors looked at quality of life in biologics. Unfortunately this is not a new topic. extensive grammar editing is required for the paper. 

In 2009, a systematic review reported that the then approved biologics (infliximab, adalimumab, certolizumab, and natalizumab) demonstrated clinical improvement in HRQoL of patients with Inflammatory bowel disease (IBD) Vogelaar L, Spijker AV, van der Woude CJ. The impact of biologics on health-related quality of life in patients with inflammatory bowel disease. Clin Exp Gastroenterol. 2009;2:101-9. Since then, despite a staggering upsurge of CD management strategies and the availability of novel biological agents, there is a scarcity of literature that could validate their efficacy by best clinical evidence. Therefore, this systematic review aimed to evaluate the outcomes of the currently approved and promising in-development biological agents and small molecule agents in improving HRQoL in patients with moderate to severe CD.

We have revised the whole manuscript for any grammar errors.

  1. the introduction needs to be shortened. there is too much information that may be redundant and as a result may sway the reader away from the main goal at hand.

Thanks for the observation. We have shortened the introduction as advised.

  1. The results section should be made more succinct.

We have reviewed the results section and found it to be accurate and precise, which corroborates well with the results of a systematic review. Some further revisions have been done to make them more succinct.

  1. Given that the data from this paper is qualitative data and not quantitative, there is for sure methodical errors that play a role and affect the results. Were you able to account for the error given that some data may be missing from the studies you extracted.

We could not understand this comment. In sharp contrast to the observation by the worthy reviewer, this systematic review is based on quantitative data and NOT qualitative data. Therefore, the comment cannot be addressed. Yes, we did have some errors in the data retrieved from the selected studies, and we have detailed these errors in the study limitations. 

Round 2

Reviewer 2 Report

The authors have improved the content of the manuscript accordingly. 

Reviewer 3 Report

.